# Cefadroxil Comparable to Cephalexin: Minimum Inhibitory Concentrations among Methicillin-Susceptible *Staphylococcus aureus* Isolates from Pediatric Musculoskeletal Infections

Andrew S. Haynes,[a,b] Andrea Prinzi,[a,c] Lori J. Silveira,[b] Sarah K. Parker,[a,b] Jed N. Lampe,[d] Jeffrey S. Kavanaugh,[e] Alexander R. Horswill,[e,f] Douglas Fish[d]

aChildren's Hospital Colorado, Department of Pediatrics, Section of Pediatric Infectious Diseases, Aurora, Colorado, USA
bUniversity of Colorado Anschutz Medical Campus, School of Medicine, Aurora, Colorado, USA
cUniversity of Colorado Anschutz Medical Campus, Graduate School, Aurora, Colorado, USA
dUniversity of Colorado Skaggs School of Pharmacy and Pharmaceutical Sciences, Aurora, Colorado, USA
eDepartment of Immunology and Microbiology, University of Colorado School of Medicine, Aurora, Colorado, USA
fDepartment of Veterans Affairs, Eastern Colorado Healthcare System, Aurora, Colorado, USA

**ABSTRACT** Cephalexin and cefadroxil are oral first-generation cephalosporins used to treat methicillin-susceptible *Staphylococcus aureus* (MSSA) infections. Despite its shorter half-life, cephalexin is more frequently prescribed, although cefadroxil is an appealing alternative, given its slower clearance and possibility for less frequent dosing. We report comparative MIC distributions for cefadroxil and cephalexin, as well as for oxacillin, cephalothin, ceftaroline, and cefazolin, for 48 unique clinical MSSA isolates from pediatric patients with musculoskeletal infections. Both cefadroxil and cephalexin had $MIC_{50}$ values of 2 $\mu$g/mL and $MIC_{90}$ values of 4 $\mu$g/mL. $MIC_{50}$s for oxacillin, cephalothin, and ceftaroline were $\leq$0.25 $\mu$g/mL, and cefazolin's $MIC_{50}$ was 0.5 $\mu$g/mL. While cefadroxil and cephalexin MICs are higher than those for other active agents, the distributions of MICs for cefadroxil and cephalexin are statistically equivalent, suggesting similar *in vitro* MSSA activities. Cefadroxil should be further considered an alternative agent to cephalexin, although additional work is needed to identify the optimal dose and frequency of these antibiotics for the treatment of serious MSSA infections.

**IMPORTANCE** Cephalexin and cefadroxil are oral antibiotics that are used to treat serious infections due to the bacteria MSSA. While cephalexin is used more commonly, cefadroxil is excreted from the body more slowly; this generally allows patients to take cefadroxil less frequently than cephalexin. In this study, we compared the abilities of cefadroxil, cephalexin, and several other representative intravenous antibiotics to inhibit the growth of MSSA in the laboratory. Bacterial samples were obtained from children with bone, joint, and/or muscle infections caused by MSSA. We found that cefadroxil and cephalexin inhibited the growth of MSSA at similar concentrations, suggesting similar antibacterial potencies. The selected intravenous antistaphylococcal antibiotics generally inhibited bacterial growth with lower antibiotic concentrations. Based on these results, cefadroxil should be further considered an alternative oral antibiotic to cephalexin, although future research is needed to identify the optimal dose and frequency of these antibiotics for serious infections.

**KEYWORDS** MIC, MSSA, cefadroxil, cephalexin, musculoskeletal infection, pediatric

Antistaphylococcal penicillins (e.g., oxacillin) and first-generation cephalosporins are the mainstays of treatment for methicillin-susceptible *Staphylococcus aureus* (MSSA) infections (1). For oral therapy, cephalexin is most commonly used, given its favorable side effect profile and low cost (1). Due to its short half-life, ~1.1 h in pediatric patients (2), cephalexin is optimally dosed four times per day (QID) for serious

Address correspondence to Andrew S. Haynes, andrew.haynes@childrenscolorado.org.

The authors declare no conflict of interest.

infections, such as musculoskeletal infections (3). However, three times per day (TID) dosing is commonly used, given concerns regarding adherence with QID dosing, with some clinical data supporting this practice (4, 5). Pharmacodynamic (PD) modeling suggests that even twice daily (BID) dosing may be sufficient in some scenarios (6).

Cefadroxil is another oral first-generation cephalosporin; it is an infrequently prescribed but appealing alternative to cephalexin based on its longer half-life, ~1.5 to 2 h in adults (7, 8). Given its slower clearance, it can likely be dosed less frequently than cephalexin, although there are insufficient pharmacokinetic (PK) or clinical data in children to determine the optimal dose and frequency for serious infections. For osteomyelitis, treatment guidelines from Europe have recommended TID cefadroxil dosing in children, which is more frequent than the FDA-approved once daily and BID dosing, although BID dosing may also be effective (3, 9–11). Despite these theoretical benefits, cefadroxil has not gained widespread use, especially in pediatrics, in part due to the paucity of pediatric PK/PD data and uncertainty about cefadroxil's range of MICs for MSSA.

According to the Clinical and Laboratory Standards Institute (CLSI), susceptibility of *S. aureus* to oxacillin or cefoxitin is used to imply susceptibility to most cephalosporins (except ceftazidime) because direct MIC testing of other cephalosporins may classify an isolate as falsely susceptible (12). Although oxacillin-susceptible isolates are considered cephalosporin susceptible, MIC distributions vary from drug to drug, and MICs for most cephalosporins are higher than oxacillin's MIC. However, knowledge of the population distribution of MICs for a given antibiotic is essential for PD modeling to understand ideal drug exposure and dosing. Therefore, we conducted a study to determine comparative MIC distributions of cefadroxil, cephalexin, and several other representative agents used for *S. aureus* infections.

## RESULTS

Overall, 49 unique MSSA isolates were identified (limited to a single isolate per patient). After exclusion of 1 isolate because of skipped wells and MIC discordance between duplicates for multiple panel antibiotics, 48 isolates obtained from blood (81%), bone (15%), and synovial fluid (4%) cultures were included in the analysis. The cefazolin MIC for 1 isolate was excluded given a 2-dilution difference between duplicates. All 6 panel antibiotics performed within reported quality control ranges for American Type Culture Collection (ATCC) *S. aureus* 29213. The source patients for the included isolates had a mean age of 9.1 years (standard deviation [SD], 4.8 years), and 52% were male ($n = 25$), with most having osteomyelitis (88%).

MIC distributions, as well as $MIC_{50}$ and $MIC_{90}$ values, for each drug are displayed in Table 1. MICs were lowest for oxacillin, cephalothin, and ceftaroline, with $MIC_{50}$ values of $\leq 0.25$ $\mu$g/mL. Cefazolin had $MIC_{50}$ and $MIC_{90}$ values of 0.5 $\mu$g/mL. Cefadroxil and cephalexin had the highest MICs of the included drugs, with $MIC_{50}$s of 2 $\mu$g/mL and $MIC_{90}$s of 4 $\mu$g/mL. Differences in MIC distributions between cefadroxil and cephalexin were compared for each isolate using the Wilcoxon signed-rank test; no significant difference between MIC distributions was found, with the median of the difference between MICs being 0 (interquartile range [IQR], 0 to 0; $P = 0.28$). When looking at isolates with a discordance between cefadroxil and cephalexin MICs, there was not a trend for one agent to have a consistently lower MIC. Among the 48 isolates, cefadroxil and cephalexin MICs were equal for 29 (60%); the cefadroxil MIC was 1 dilution lower for 10 (21%), and the cephalexin MIC was 1 dilution lower for 9 (19%). Table 2 displays the correlation between cefadroxil and cephalexin MICs.

## DISCUSSION

While MICs of cefadroxil and cephalexin for MSSA are both higher than those of other active agents, the distributions of MICs for cefadroxil and cephalexin are statistically equivalent, suggesting that the two drugs have similar degrees of *in vitro* activity. These findings are similar to prior reports. $MIC_{90}$ values for cephalexin and cefadroxil vary across published studies, from 4 $\mu$g/mL to 8 $\mu$g/mL, and up to 16 $\mu$g/mL, for both

**TABLE 1** Number of clinical MSSA isolates at each MIC, according to antibiotic

| MIC ($\mu$g/mL) | No. of isolates with indicated MIC for: | | | | | |
|---|---|---|---|---|---|---|
| | Oxacillin | Cephalothin | Ceftaroline | Cefazolin | Cefadroxil | Cephalexin |
| ≤0.25 | 43 | 42 | 46 | 22 | 0 | 0 |
| 0.5 | 5 | 5 | 2 | 23 | 0 | 0 |
| 1 | 0 | 1 | 0 | 2 | 6 | 10 |
| 2 | 0 | 0 | 0 | 0 | 36 | 25 |
| 4 | 0 | 0 | 0 | 0 | 5 | 13 |
| 8 | 0 | 0 | 0 | 0 | 1 | 0 |
| 16 | 0 | 0 | 0 | 0 | 0 | 0 |
| Total | 48 | 48 | 48 | 47 | 48 | 48 |
| $MIC_{50}$ | ≤0.25 | ≤0.25 | ≤0.25 | 0.5 | 2 | 2 |
| $MIC_{90}$ | 0.5 | 0.5 | ≤0.25 | 0.5 | 4 | 4 |

drugs (13–16), although no prior study included exclusively strains causing osteoarticular infections. In the two studies that evaluated cephalexin alongside cefadroxil, $MIC_{90}$ values for cefadroxil were 1 dilution lower than those for cephalexin (either 4 versus 8 $\mu$g/mL or 8 versus 16 $\mu$g/mL) among 20 (14) and 141 (13) strains.

*In vivo*, PK/PD target attainment and clinical success also depend on dose, absorption, half-life, and dosing interval, among other PK/PD considerations. Given the limited data available, the concentrations needed for adequate time above our described MICs (T>MIC) are likely achievable with commonly used oral dosing strategies. In a pediatric cephalexin PK study using median dosing of 120 mg/kg/day divided into TID doses, all 11 patients achieved the specified PD target of T>MIC for >40% of the dosing interval for lower MICs, but 20% of the patients failed to achieve this PD target for our described $MIC_{90}$ (4 mg/L). This finding suggests that either higher daily doses or QID dosing may be needed for some patients. Cefadroxil PK data in children are more limited. A 1978 study of 30 children receiving cefadroxil at 10 to 15 mg/kg/dose (which may not be reflective of the currently recommended doses of 75 to 150 mg/kg/day for osteomyelitis [3, 10, 11]) found peak concentrations of up to 13.7 ± 0.97 mg/L and half-lives of 1.3 to 1.5 h (17). Cefadroxil was well tolerated at these low doses. For higher cefadroxil doses, PK data are available from a single patient (18), which suggests that cefadroxil at 120 mg/kg/day, using an MIC of 8 mg/L, would achieve T>MIC of >40% of the dosing interval with BID dosing and T>MIC of >60% of the interval with TID dosing (3, 18).

This study has some limitations. Because the clinical MSSA isolates were all obtained from pediatric patients from a single institution, they may not reflect MSSA populations in other regions that may have different MIC distributions. While all included clinical isolates were MSSA, as defined by their oxacillin MICs, the clinical relevance and validity of the MIC ranges for other study antibiotics are not known. Lastly, these results reflect MIC evaluations performed with a standard organism inoculum. Some MSSA isolates exhibit elevated MICs for first-generation cephalosporins (cefazolin and cephalexin and presumably cefadroxil) when a high organism inoculum is used (4).

These data confirm that cefadroxil has similar *in vitro* activity against MSSA and should be further considered an alternative agent to cephalexin. More formal PK/PD

**TABLE 2** Comparative MICs for cefadroxil and cephalexin for 48 unique clinical isolates of MSSA

| Cephalexin MIC ($\mu$g/mL) | No. of isolates with cefadroxil MIC of: | | | | |
|---|---|---|---|---|---|
| | 1 $\mu$g/mL | 2 $\mu$g/mL | 4 $\mu$g/mL | 8 $\mu$g/mL | Total |
| 1 | 4 | 6 | 0 | 0 | 10 |
| 2 | 1 | 22 | 2 | 0 | 25 |
| 4 | 1 | 8 | 3 | 1 | 13 |
| 8 | 0 | 0 | 0 | 0 | 0 |
| Total | 6 | 36 | 5 | 1 | 48 |

evaluations and/or clinical data are necessary to confirm that cefadroxil can be used reliably for severe MSSA infections and to identify optimal dosing strategies.

## MATERIALS AND METHODS

Patients admitted to Children's Hospital Colorado between 1 January 2018 and 30 May 2020 with a musculoskeletal infection due to MSSA were eligible for inclusion. MSSA isolates were selected from primary culture media at 18 to 24 h of growth and frozen at $-20°C$ until antimicrobial susceptibility testing (AST). This study was approved by the Colorado Multiple Institutional Review Board.

Clinical isolates, as well as ATCC strain *S. aureus* 29213 for quality control, were subcultured on tryptic soy agar with 5% sheep blood (Thermo Fisher Scientific, Waltham, MA). After 18 to 24 h of incubation, isolates were subcultured on secondary blood agar plates. Overnight growth of isolates was suspended in demineralized water to an acceptable McFarland standard turbidity (a McFarland standard range of 0.08 to 0.13 with a 625-nm-wavelength GENESYS 30 spectrophotometer), and then 50 $\mu$L of each sample was transferred to 11 mL of sterile Mueller-Hinton broth (Thermo Fisher Scientific). From this, 50 $\mu$L was inoculated into each well of a 96-well customized Sensititre panel for AST, including oxacillin, cephalothin, ceftaroline, cefazolin, cefadroxil, and cephalexin in dilutions ranging from 0.25 to 16 $\mu$g/mL (Thermo Fisher Scientific). Panels were sealed and incubated for 18 to 24 h in a 35 to 37°C non-$CO_2$ incubator. All panels contained duplicate wells at each antibiotic concentration, and two sets of MICs were documented for each isolate. According to CLSI recommendations, the MIC for each isolate was considered the first well with no visible growth to the unaided eye. When comparing duplicate results with different MICs, the most conservative (highest) MIC was taken, and isolates with duplicate MICs differing by more than 1 dilution were excluded from analysis. The highest MIC was accepted for skipped wells (i.e., wells with visible growth in the wells below and above) if only one well was skipped, while data were excluded for antibiotics with more than one skipped well.

Summary statistics were used to describe patients and isolate data. The Wilcoxon signed-rank test was used to compare MIC distributions between cefadroxil and cephalexin, and IQRs and $P$ values were obtained. Statistical analyses were performed using R (version 4.1.2; R foundation, Vienna, Austria, 2021) and SAS v9.4 (SAS Institute Inc., Cary, NC) software.

## ACKNOWLEDGMENTS

This research received no specific grant from any funding agency in the public, commercial, or not-for-profit sectors. A.R.H. was supported by U.S. Department of Veterans Affairs Merit Award I01 BX002711.

We thank the Children's Hospital Colorado microbiology laboratory for assistance in collecting the clinical isolates used in this study.

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
