## [Reviewer comments · Microbiology Spectrum]

Microbiology Spectrum

Cefadroxil comparable to cephalexin: minimum inhibitory concentrations among methicillin-susceptible *Staphylococcus aureus* isolates from pediatric musculoskeletal infections

Andrew Haynes, Andrea Prinzi, Lori Silveira, Sarah Parker, Jed Lampe, Jeffrey Kavanaugh, Alexander Horswill, and Douglas Fish

Corresponding Author(s): Andrew Haynes, University of Colorado Anschutz Medical Campus

Review Timeline:

Submission Date:	March 31, 2022
Editorial Decision:	May 5, 2022
Revision Received:	May 23, 2022
Accepted:	May 25, 2022

Editor: Wendy Szymczak

Reviewer(s): Disclosure of reviewer identity is with reference to reviewer comments included in decision letter(s). The following individuals involved in review of your submission have agreed to reveal their identity: Jonathon Chase McNeil (Reviewer #1); Anita Campbell (Reviewer #2)

Transaction Report:

DOI: <https://doi.org/10.1128/spectrum.01039-22>

May 5, 2022

Dr. Andrew S Haynes
University of Colorado Anschutz Medical Campus
Aurora, CO

Re: Spectrum01039-22 (Cefadroxil comparable to cephalexin: minimum inhibitory concentrations among methicillin-susceptible *Staphylococcus aureus* isolates from pediatric musculoskeletal infections)

Dear Dr. Andrew S Haynes:

Link Not Available

Sincerely,

Wendy Szymczak

Journals Department
Reviewer comments:

Reviewer #1 (Comments for the Author):

Dr. Haynes and Colleagues present a brief report describing MICs to antistaphylococcal beta-lactams among pediatric MSSA musculoskeletal infection isolates. The overall object of this study is to evaluate the relative in vitro susceptibility of isolates to cefadroxil compared to other first generation cephalosporins (oxacillin and ceftaroline were also measured). While cephalexin is commonly used, use of cefadroxil may allow for fewer daily dose administrations. The paper does provide some information of clinical interest and the microbiologic methods appear sound. Appropriate controls were used. The paper would be improved by putting the results into context a bit more. It is difficult to appreciate the value of a given MIC without understanding what are the expected serum / tissue levels of drug.

My specific queries / comments / suggestions are listed below:

1. Introduction. The authors comment that some experts recommend thrice daily cephalexin rather than four times daily out of concerns for adherence. Of note, in two recent publications, MSSA osteoarticular infections were treated with three times daily cephalexin with high levels of treatment success (Ramchandrar N, et al. 2020. *Pediatr Infect Dis J* 39:523-525; McNeil et al. *Antimicrob Agents Chemother*. 2020; 64: e00703) and in one of these papers tid dosing was comparable to qid dosing in terms of efficacy (albeit uncontrolled and sample size limiting).

2. Introduction. The authors write, "While cephalexin is typically dosed 3-4 times per day, cefadroxil can likely be dosed 2-3 times per day" and reference the package insert. Some expansion on this is probably warranted. Cefadroxil is approved for treatment of UTI, SSTI and streptococcal pharyngitis in children with daily or bid dosing. Why would three times daily dosing of cefadroxil be needed?

3. Methods. The authors should clarify that only a single isolate was used per clinical case (for example, a blood culture and bone aspirate were not used from the same patient). My assumption from the paper was that each isolate was truly unique although this should be made explicit.

4. Results. The MIC data as presented in the table are interesting and the methods appear rigorous. In the text, may consider reporting the MIC50 and MIC90 as this is the more common practice rather than median and IQR.

5. Discussion. As stated above, can the authors put the MIC findings in the context of expected serum levels of cephalexin/cefadroxil reported in the literature? This would help at better appreciating their results.

Reviewer #2 (Comments for the Author):

Cefadroxil comparable to cephalexin: minimum inhibitory concentrations among methicillin- susceptible *Staphylococcus aureus* isolates from pediatric musculoskeletal infections

This manuscript describing the in vitro testing results of 48 MSSA isolates from paediatric bone and joint samples is well written and has some interesting findings. Although not entirely new as the MICs of cephalosporins versus cefadroxil have been reported in paper 9 and 10 referenced, this paper revisits a topic that has largely not been discussed in the recent literature and the angle of the findings suggesting there may be a role in paediatrics and more PK/PD and clinical studies required.

Scientific quality: The methods are well described, and limitations clearly outlined.

Relevance to journal/reader: I think this paper is of interest and relevant to the readership and encouragement for more paediatric PK/PD data for drugs such of this is important as children are frequently a forgotten cohort.

General comments:

Introduction

Line 62 Suggest referencing the sentence "However, three-times daily (TID) dosing is commonly used given concerns regarding adherence with QID dosing" could consider the recent publication

https://journals.lww.com/pidj/Fulltext/2020/06000/ Twice__and_Thrice_daily_Cephalexin_Dosing_for.12.aspx

Line 66 "cefadroxil is an appealing alternative to cephalexin based on its longer half-life, ~1.5-2 hours in adults (4, 5). While cephalexin is typically dosed 3-4 times per day, cefadroxil can likely be dosed 2-3 times per day (6)." Is this the dosing interval of 2-3 times per day recommended in adults or children or both, recommend specifying?

Line 59 "For oral therapy, cephalexin is most commonly used given its favorable side effect profile and low cost." Is there a reference for this at all and are you referring to in children (I think this is commonly the case in paediatrics but thought that in adults oxacillin is more commonly favoured but again jurisdiction dependent)?

For the below comments in the introduction and discussion respectively:

Line 68 "Despite these theoretical benefits, cefadroxil has not gained widespread use, especially in pediatrics, in part due to the paucity of pediatric pharmacokinetic and pharmacodynamic (PK/PD) data and uncertainty about cefadroxil's range of minimum inhibitory concentrations

line 144 "Given the limited data available, the concentrations needed for adequate time above our described MICs are likely achievable with commonly used oral dosing strategies"

Is there any PK/PD data in children (even down to 12) for cefadroxil? If so it would be helpful for the reader to outline this data in the manuscript. What about palatability of cefadroxil given your angle on the data in the paediatric context? Are there any safety

profile differences?

Line 139-141 "These findings are similar to prior reports. Cephalixin MIC90 values for MSSA of 4 µg/mL (8) and 8 µg/mL (9, 10) have been reported. Previously reported cefadroxil MIC90 values have been 4 µg/mL (9, 10)." how many MSSA isolates were tested in reference 9 and 10 would be helpful for the reader?

The results, discussion and conclusions are sound and the authors have appropriately mentioned the importance of more PK/PD data and clinical studies in their conclusion.

Staff Comments:

Preparing Revision Guidelines

Please return the manuscript within 60 days; if you cannot complete the modification within this time period, please contact me. If you do not wish to modify the manuscript and prefer to submit it to another journal, please notify me of your decision immediately so that the manuscript may be formally withdrawn from consideration by Microbiology Spectrum.

Manuscript review 05/05/2022 AAC

Cefadroxil comparable to cephalexin: minimum inhibitory concentrations among methicillin-susceptible *Staphylococcus aureus* isolates from paediatric musculoskeletal infections

This manuscript describing the *in vitro* testing results of 48 MSSA isolates from paediatric bone and joint samples is well written and has some interesting findings. Although not entirely new as the MICs of cephazolin versus cefadroxil have been reported in paper 9 and 10 referenced, this paper revisits a topic that has largely not been discussed in the recent literature and the angle of the findings suggesting there may be a role in paediatrics and more PK/PD and clinical studies required.

Scientific quality: The methods are well described, and limitations clearly outlined.

Relevance to journal/reader: I think this paper is of interest and relevant to the readership and encouragement for more paediatric PK/PD data for drugs such of this is important as children are frequently a forgotten cohort.

General comments:

Introduction

Line 62 Suggest referencing the sentence “However, three-times daily (TID) dosing is commonly used given concerns regarding adherence with QID dosing” could consider the recent publication

https://journals.lww.com/pidj/Fulltext/2020/06000/ Twice_and_Thrice_daily_Cephalexin_Dosing_for.12.aspx

Line 66 “cefadroxil is an appealing alternative to cephalexin based on its longer half-life, ~1.5-2 hours in adults (4, 5). While cephalexin is typically dosed 3-4 times per day, cefadroxil can likely be dosed 2-3 times per day (6).” Is this the dosing interval of 2-3 times per day recommended in adults or children or both, recommend specifying?

Line 59 “For oral therapy, cephalexin is most commonly used given its favorable side effect profile and low cost.” Is there a reference for this at all and are you referring to in children (I think this is commonly the case in paediatrics but thought that in adults oxacillin is more commonly favoured but again jurisdiction dependent)?

For the below comments in the introduction and discussion respectively:

Line 68 “Despite these theoretical benefits, cefadroxil has not gained widespread use, especially in pediatrics, in part due to the paucity of pediatric pharmacokinetic and pharmacodynamic (PK/PD) data and uncertainty about cefadroxil’s range of minimum inhibitory concentrations

line 144 “Given the limited data available, the concentrations needed for adequate time above our described MICs are likely achievable with commonly used oral dosing strategies”

Is there any PK/PD data in children (even down to 12) for cefadroxil? If so it would be helpful for the reader to outline this data in the manuscript. What about palatability of cefadroxil given your angle on the data in the paediatric context? Are there any safety profile differences?

Line 139-141 “These findings are similar to prior reports. Cephalexin MIC90 values for MSSA of 4 µg/mL (8) and 8 µg/mL (9, 10) have been reported. Previously reported cefadroxil MIC90 values have been 4 µg/mL (9, 10).” how many MSSA isolates were tested in reference 9 and 10 would be helpful for the reader?

The results, discussion and conclusions are sound and the authors have appropriately mentioned the importance of more PK/PD data and clinical studies in their conclusion.

5/23/2022

Response to Reviewers – Cefadroxil comparable to cephalixin: minimum inhibitory concentrations among methicillin-susceptible *Staphylococcus aureus* isolates from pediatric musculoskeletal infections

Corresponding Author: Andrew Haynes

Reviewer #1 (Comments for the Author):

Dr. Haynes and Colleagues present a brief report describing MICs to antistaphylococcal beta-lactams among pediatric MSSA musculoskeletal infection isolates. The overall object of this study is to evaluate the relative in vitro susceptibility of isolates to cefadroxil compared to other first generation cephalosporins (oxacillin and ceftaroline were also measured). While cephalixin is commonly used, use of cefadroxil may allow for fewer daily dose administrations. The paper does provide some information of clinical interest and the microbiologic methods appear sound. Appropriate controls were used. The paper would be improved by putting the results into context a bit more. It is difficult to appreciate the value of a given MIC without understanding what are the expected serum / tissue levels of drug.

My specific queries / comments / suggestions are listed below:

1. Introduction. The authors comment that some experts recommend thrice daily cephalixin rather than four times daily out of concerns for adherence. Of note, in two recent publications, MSSA osteoarticular infections were treated with three times daily cephalixin with high levels of treatment success (Ramchandrar N, et al. 2020. *Pediatr Infect Dis J* 39:523-525; McNeil et al. *Antimicrob Agents Chemother*. 2020; 64: e00703) and in one of these papers tid dosing was comparable to qid dosing in terms of efficacy (albeit uncontrolled and sample size limiting).

Response: Added references to these two papers as supporting clinical data.

2. Introduction. The authors write, "While cephalixin is typically dosed 3-4 times per day, cefadroxil can likely be dosed 2-3 times per day" and reference the package insert. Some expansion on this is probably warranted. Cefadroxil is approved for treatment of UTI, SSTI and streptococcal pharyngitis in children with daily or bid dosing. Why would three times daily dosing of cefadroxil be needed?

Response: The rationale for possibly needing TID dosing in osteomyelitis is to achieve higher T>MIC targets, and with higher probability of target attainment, given the seriousness of the infection, especially given the need for bone penetration. Cefadroxil is used for osteoarticular infections more commonly outside of the United States (where we are based). Cefadroxil is included in multiple treatment guidelines from Europe for osteomyelitis. In those guidelines, it is frequently dosed much higher than the FDA approved doses for UTI, SSTI, GAS pharyngitis, etc. Spanish guidelines (from Sociedad Espanola de Infectologia Pediatrica) recommend 90 mg/kg/day divided q8h, the French Pediatric Infectious Disease Group recommends 150 mg/kg/day in 3 divided doses, and the European Society for Pediatric Infectious Diseases (ESPID) recommends 75-150 mg/kg/day, in 3-4 doses. In line with these recommendations, a review paper on the treatment of pediatric osteomyelitis by DeRonde et al suggests 120mg/kg/day q8h. We have expanded the discussion of this in the manuscript to reflect this data and have included some of the above references.

3. Methods. The authors should clarify that only a single isolate was used per clinical case (for example, a blood culture and bone aspirate were not used from the same patient). My assumption from the paper was that each isolate was truly unique although this should be made explicit.

Response: This is correct. The 49 unique isolates were obtained from 49 different patients, with no duplication of isolates. Added clarification in the text.

4. Results. The MIC data as presented in the table are interesting and the methods appear rigorous. In the text, may consider reporting the MIC50 and MIC90 as this is the more common practice rather than median and IQR.

Response: Updated to include MIC50/90 values in the text in place of median/IQR. Also updated the abstract similarly.

5. Discussion. As stated above, can the authors put the MIC findings in the context of expected serum levels of cephalexin/cefadroxil reported in the literature? This would help at better appreciating their results.

Response: Added paragraph to the discussion presenting available PK/PD data for cephalexin and cefadroxil to give the reader a general sense of T>MIC that might be achieved with various dosing strategies and MIC values.

Reviewer #2 (Comments for the Author):

Cefadroxil comparable to cephalexin: minimum inhibitory concentrations among methicillin- susceptible Staphylococcus aureus isolates from pediatric musculoskeletal infections

This manuscript describing the in vitro testing results of 48 MSSA isolates from paediatric bone and joint samples is well written and has some interesting findings. Although not entirely new as the MICs of cephalosporins versus cefadroxil have been reported in paper 9 and 10 referenced, this paper revisits a topic that has largely not been discussed in the recent literature and the angle of the findings suggesting there may be a role in paediatrics and more PK/PD and clinical studies required.

Scientific quality: The methods are well described, and limitations clearly outlined.

Relevance to journal/reader: I think this paper is of interest and relevant to the readership and encouragement for more paediatric PK/PD data for drugs such of this is important as children are frequently a forgotten cohort.

General comments:

Introduction

Line 62 Suggest referencing the sentence "However, three-times daily (TID) dosing is commonly used given concerns regarding adherence with QID dosing" could consider the recent publication https://journals.lww.com/pidj/Fulltext/2020/06000/ Twice__and_Thrice_daily_Cephalexin_Dosing_for.12.aspx

Response: Added this reference to the manuscript.

Line 66 "cefadroxil is an appealing alternative to cephalexin based on its longer half-life, ~1.5-2 hours in adults (4, 5). While cephalexin is typically dosed 3-4 times per day, cefadroxil can likely be dosed 2-3 times per day (6)." Is this the dosing interval of 2-3 times per day recommended in adults or children or both, recommend specifying?

Response: Clarified that this dosing is recommended in children in various osteomyelitis guidelines (see response to Reviewer #1 as well, and citations added to manuscript). FDA approval for cefadroxil includes only once daily and twice daily dosing. Because oral step-down therapy is used less commonly in the treatment of osteomyelitis in adults, there is less published on the subject. Anecdotally, when we've heard of cefadroxil being used in adults for osteomyelitis, dosing of 500-1000mg BID is generally used.

Line 59 "For oral therapy, cephalexin is most commonly used given its favorable side effect profile and low cost." Is there a reference for this at all and are you referring to in children (I think this is commonly the case in paediatrics but thought that in adults oxacillin is more commonly favoured but again jurisdiction dependent)?

Response: This may be location dependent. As far as we are aware, the only oral anti-staphylococcal penicillin available in the United States (where we are located) is dicloxacillin. This is not available as a suspension, which limits its use in children. It also has more GI intolerance and higher rates of protein binding than cephalexin, so in our experience, it is almost never used for enteral therapy over cephalexin. In the new IDSA pediatric acute hematogenous osteomyelitis guidelines, no oral anti-staphylococcal penicillin is mentioned—in fact, cephalexin is the only oral anti-staphylococcal beta-lactam mentioned (no other oral cephalosporins are mentioned, including cefadroxil). We have added the IDSA guideline as a reference, though acknowledge that this only applies to pediatrics.

For the below comments in the introduction and discussion respectively:

Line 68 "Despite these theoretical benefits, cefadroxil has not gained widespread use, especially in pediatrics, in part due to the paucity of pediatric pharmacokinetic and pharmacodynamic (PK/PD) data and uncertainty about cefadroxil's range of minimum inhibitory concentrations

line 144 "Given the limited data available, the concentrations needed for adequate time above our described MICs are likely achievable with commonly used oral dosing strategies"

Is there any PK/PD data in children (even down to 12) for cefadroxil? If so it would be helpful for the reader to outline this data in the manuscript. What about palatability of cefadroxil given your angle on the data in the paediatric context? Are there any safety profile differences?

Response: Added a brief discussion of available PK/PD data in cefadroxil. Anecdotally, their palatability is similar in our personal experience. There is no apparent difference in their safety profiles, with the caveat that there no published data on the use of higher doses of cefadroxil in pediatrics (other than a case report of a single patient that is referenced). That being said, its inclusion at high doses (up to 150 mg/kg/day) in multiple European treatment guidelines (as discussed in response to a similar question from Reviewer #1 above) would suggest there are no obvious safety signals using higher doses in children.

Line 139-141 "These findings are similar to prior reports. Cephalexin MIC90 values for MSSA of 4 µg/mL (8) and 8 µg/mL (9, 10) have been reported. Previously reported cefadroxil MIC90 values have been 4 µg/mL (9, 10)." how many MSSA isolates were tested in reference 9 and 10 would be helpful for the reader?

Response: We have added this detail about two of the studies, and also expanded some details about prior MIC reports for these drugs.

The results, discussion and conclusions are sound and the authors have appropriately mentioned the importance of more PK/PD data and clinical studies in their conclusion.

May 25, 2022

Dr. Andrew S Haynes
University of Colorado Anschutz Medical Campus
Aurora, CO

Re: Spectrum01039-22R1 (Cefadroxil comparable to cephalexin: minimum inhibitory concentrations among methicillin-susceptible *Staphylococcus aureus* isolates from pediatric musculoskeletal infections)

Dear Dr. Andrew S Haynes:

Your manuscript has been accepted, and I am forwarding it to the ASM Journals Department for publication. You will be notified when your proofs are ready to be viewed.

Sincerely,

Wendy Szymczak
Editor, Microbiology Spectrum
